# *Microcystis* Bloom in an Urban Lake after River Water Diversion—A Case Study

**Xiaoyan Chen [1,†], Dong Bai [1,2,†], Chunlei Song [1], Yiyong Zhou [1] and Xiuyun Cao [1,*]**

[1]   Key Laboratory of Algal Biology, State Key Laboratory of Freshwater Ecology and Biotechnology, Institute of Hydrobiology, Chinese Academy of Sciences, Wuhan 430072, China; chenxy839@ihb.ac.cn (X.C.); baidong@ihb.ac.cn (D.B.); clsong@ihb.ac.cn (C.S.); zhouyy@ihb.ac.cn (Y.Z.)

[2]   University of Chinese Academy of Sciences, Beijing 100039, China

*   Correspondence: caoxy@ihb.ac.cn; Tel./Fax: +86-27-6878-0221 (ext. 0123)

†   Xiaoyan Chen and Dong Bai are co-first authors of this article.

**Abstract:** To improve the water quality of Lake Yuehu, a water diversion from the Han River was conducted in July 2008. However, an unexpected *Microcystis* bloom occurred in the lake after water introduction. Water and sediment samples were collected from Lake Yuehu and the variation of chemical and biochemical parameters, as well as the phytoplankton community, were analyzed during the water diversion to assess its effect and to clarify the mechanism leading to the *Microcystis* bloom. The nitrogen (N) concentration was increased and phosphorus (P) concentration decreased in Lake Yuehu after receiving water from the Han River, which had a high loading of N and a low loading of P. These conditions may benefit the growth and dominance of non-$N_2$ fixing *Microcystis*, as it may not have suffered from P limitation during our study because it did not produce extracellular phosphatase, which worked as an indicator of P deficiency, as evidenced by the in situ enzyme-labelled fluorescence. Notably, the sediment Fe (OOH)~P content significantly decreased in Lake Yuehu; this pulsed release of P from the sediment might have sustained the *Microcystis* bloom. Based on our results, algal blooms may occur as a consequence of conducting water diversion projects to improve water quality.

**Keywords:** water transfer; nutrient variation; cyanobacterial bloom; sediment p fraction; alkaline phosphatase activity

## 1. Introduction

Eutrophication and cyanobacterial blooms caused by genera such as *Microcystis* have become increasingly common in freshwater ecosystems globally, especially in urban lakes, resulting in a serious threat to recreational functions [1–3]. Most *Microcystis* strains have been demonstrated to produce potent hepatatoxin microcystin, blooms of which would threaten human health, recreational activities and fisheries. *Microcystis* blooms have been recorded in at least 108 countries and 79 of them have reported the hepatatoxin microcystin, illustrating the massive expansion of harmful *Microcystis* all over the world [4]. It is widely accepted that high nutrient loading, increasing temperatures and decreasing flow rates that create quiescent or stagnant waters are the main triggers that cause the blooms [5–7]. Several approaches to suppress cyanobacterial blooms or to minimize the effects of eutrophication have been developed such as mechanical mixing, flushing, the application of algaecides and biomanipulation [7–10]. Diverting water from rivers to lakes is one of the approaches that targets a reduction in the incidence of blooms or eutrophication in lakes. Nevertheless, it is unclear whether water diversion could improve water quality, and little is known about the influence mechanism of water diversion on blooms of *Microcystis* in lakes.

Water diversion might positively affect the lakes concerned by diluting the concentration of nutrients such as nitrogen (N) and phosphorus (P) [11–13] and causing disturbance to remove stratification, which restrains the growth of cyanobacteria and alleviates cyanobacterial blooms in the short-term [14–16]. For example, in China, water was diverted from the Yangtze River into Lake Taihu from 2002 to 2003, contributing to a temporary deduction of the total N (TN), total P (TP) and chlorophyll in many areas of the lake [17]. Furthermore, in Moses Lake in the USA, the introduction of diluting water reduced the algal biomass and improved the water quality by reducing the nutrient concentration [18]. Therefore, water diversion has become an effective remedy in arresting and reducing bloom intensities, at least in the short term. However, negative impacts of water diversion on lakes were also observed. Firstly, water transfer might introduce excessive nutrients or sediment into the receiving waterbody, increasing the risk of eutrophication. For example, the water diversion project from the Yangtze River resulted in a higher nitrate loading in Lake Dazong of China by studying [19]. Secondly, influent water with pulsed nutrients would induce the transition of phytoplankton community and alter the predominant species, potentially facilitating a harmful algal bloom (especially of toxin cyanobacteria) [20,21]. Accumulations of blue-green algae occurred after when water was diverted into Lake Pontchartrain from the Mississippi River in America [22]. In addition, although water transfer might dilute nutrient concentrations, dilution would cause larger diffusive gradients between the sediment and overlying water and enhance internal P release [23]. On the other hand, there was no significant variation in the alkaline phosphatase activity (APA) or kinetic parameters of Lake Taihu during the diversion, indicating that little P concentration variation occurred [24,25]. Therefore, views on the effect of water diversion in terms of cyanobacterial blooms and eutrophication vary widely. More case studies to monitor the process and underlying mechanism of water diversion are highly required.

Lake Yuehu, a central urban lake, has suffered from serious anthropogenic pollution since the late 1990s [26,27]. From June to August in 2008, a water diversion was used to alleviate the odorous situation in the lake. It is a representative case for studying and evaluating the effects of a water diversion project on urban landscape lakes.

The objective of this study was to assess the impact of water diversion from a river on an urban lake, paying special attention to nutrient variation and phytoplankton community shift as well as the phytoplankton physiological response. We collected water, phytoplankton and sediment samples from Lake Yuehu and the Han River and analyzed the concentration of different forms of N and P, and APA in the water column and sediment. Phytoplankton extracellular alkaline phosphatase was detected at the single cell level during water diversion. Our results provide insights for lake eutrophication management when considering water diversion.

## 2. Materials and Methods

Lake Yuehu (30°33′39.48″ N, 114°14′59.98″ E) is located at Wuhan of Hubei province in central China in the alluvial plain of the Yangtze River. It is adjacent to the Han River to the north and connected with the river by brakes. The Han River flows eastward into the Yangtze river and is its largest tributary (Figure 1), with a length of 1577 km, tributary area of 1,590,000 km$^2$ and 245 hundred million m$^3$ of mean annual runoff. Lake Yue functions as an urban lake with a total area of 0.66 km$^2$ and an average depth of 1.2 m. Its surface water temperature ranges from 6 °C in winter to 30 °C in summer. The average annual rainfall is 1269 mm, which occurs mostly from June to August.

Water diversion from the Han River to Lake Yuehu was conducted from June to July 2008, during which time the air temperature varied between 28–35 °C and no sustained wind occurred. On 28 June 2008, 1,000,000 m$^3$ of the water from Lake Yuehu was released into the Han River by opening the brakes connecting with the lake and the river, whose water level was low at that time. After that, the lake's sediment was exposed to the air in order to sterilize and deodorize the mud. When the water level of the Han River rose, it was then transferred into Lake Yuehu on 26 July. The water introduction processed from 26 July to 31 July, and 800,000 m$^3$ of water flowed into the lake, with the aim of improving its water quality.

Three sampling sites in the lake and one sampling site in the river were established (Figure 1). Surface water (0–50 cm) was collected on 14 July, 28 July, and 1 August both in Lake Yuehu and the Han River, and on 7, 14, 21 and 28 August in Lake Yuehu. Samples were used to determine the structure and cell density of phytoplankton, and concentrations of TP, total dissolved P (DTP), soluble reactive P (SRP), TN, ammonium ($NH_4^+$-N), nitrite ($NO_2^-$-N), and nitrate ($NO_3^-$-N). APA analysis was conducted in Lake Yuehu only. Since the introduction of river water into the lake took place from 26 to 31 July 2008, we defined the period from 24 June to 14 July as the period before water diversion, while 26 July to 28 August was considered the period after water diversion.

Surface sediments (0–10 cm) were sampled using a Peterson grab sampler to analyze P fractionation and kinetics of alkaline phosphatase in sediment.

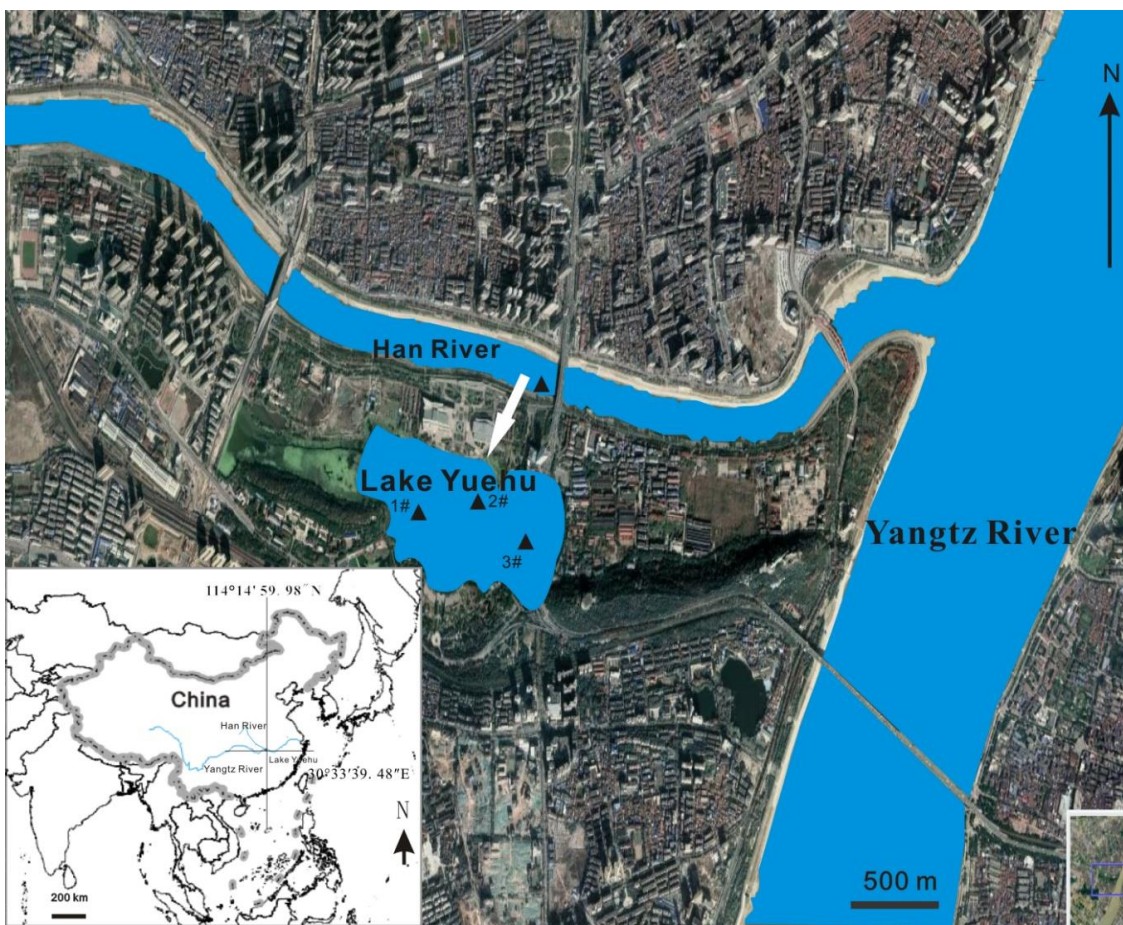

**Figure 1.** Location of Han River and Lake Yuehu. Triangles indicate the sampling sites.

*2.1. Chemical Analysis*

The SRP measurement was performed in triplicate according to the molybdate blue method [28]. Samples were processed by filtering through 0.45 μm (Mosu Science Equipment Corporation, Shanghai, China) filters. Then, 1.6 mL color agent was added to 8 mL filtrate and a standard phosphate gradient solution, which was diluted to 10 mL for measurements of phosphate. The color agent was a mixed reagent consists which consisted of an acidified solution of ammonium molybdate containing ascorbic acid and a small amount of antimony. The value under the spectrophotometer (Modern Science Corporation, Shanghai, China) was read at 882 nm after incubating 15 min of incubation at room temperature. TP and DTP concentrations were measured following digestion according to the method reported by Beattie et al. [29]. Triplicate water samples from the field for measurement of the TP concentration and the filtrate for measurement of the DTP concentration were digested by potassium peroxodisulfate; then, the phosphate concentrations were measured by the blue molybdate method

(the same as was used for SRP). Particulate P (PP) and dissolved organic P (DOP) were calculated as PP = TP − DTP and DOP = DTP − SRP, respectively. Concentrations of different forms of N ($NH_4^+$-N, $NO_2^-$-N and $NO_3^-$-N) were measured using the procedure provided by the standard methods [30]. A triplicate of 5 mL filtered water samples was mixed with 3 mL phenol solution and sodium hypochlorite for measurement of ammonium, and 3 mL p-aminobenzene sulfanilic acid and 1-naphthylamine solution for measurement of nitrite. These samples were then diluted to 25 mL. The absorbance of the solution was read at 625 nm for ammonium and 530 nm for nitrite after 40 min of incubation at room temperature. The nitrate concentration was determined by reading the absorbance of filtered water samples at 220 nm and 275 nm after purifying with carbon powder. Dissolved inorganic of N (DIN) was calculated by the sum of $NH^{4+}$-N, $NO^{2−}$-N and $NO^{3−}$-N.

### 2.2. Alkaline Phosphatase Activities (APA) and the Kinetics

#### 2.2.1. APA in the Water Column

APA in the water column and the kinetics of alkaline phosphatase in sediment were assayed by p-nitrophenylphosphate (pNPP), which was hydrolyzed by the enzyme to yield p-nitrophenol. Water samples were mixed in Tris buffer (pH 8.9). The pNPP was added to triplicate water samples at a final pNPP concentration of 0.3 mmol·L$^{-1}$. Samples were incubated at 37 °C for 1 h, then 1.6 mL of samples were centrifuged for 10 min at 3000 rpm. A total of 4 mL 0.1 M NaOH was used in 1 mL supernatant to stop the reaction. APA in the water column was indicated by an increase in light absorbance at 410 nm using a spectrophotometer [31].

#### 2.2.2. Kinetics of Alkaline Phosphatase in Sediment

The pNPP was added to slurries in triplicate sediment samples at six final concentrations ranging from 0.0625 to 1.0 mmol·L$^{-1}$ to measure the parameters of kinetics of alkaline phosphatase in sediment. The steps were the same as those to measure the APA in the water column. APA was converted to absolute units using a standard curve based on enzymatically hydrolyzed p-nitrophenol. The parameters of the Michaelis constant ($K_m$) which characterizes the affinity of enzyme for substrate and the maximum velocity of the enzyme ($V_{max}$) which characterizes the catalyzed rate of enzyme were estimated by fitting the linearized Michaelis–Menten equation [32].

#### 2.2.3. APA on Single-Cell Level

Extracellular alkaline phosphatase of phytoplankton on single-cell was detected using the enzyme-labeled fluorescence (ELF) method. Duplicate water samples with algae obtained from field were treated using ELF® 97 phosphate (ELFP, InvitrogenTM, Life Technologies Corporation, Eugene, OR, USA) on the basis of the protocol described in Strojsova et al. [33]. In the amount of 0.5 mL, incubations were started by adding the ELFP solution (final concentration 27 μM) and samples were incubated at 25 °C for 2.5 h. Each incubation was terminated by transferring the sample to a filter holder (diameter 7 mm) with a membrane filter (Millipore; 0.22 μm pore size, Millipore Corporation, MA, USA). The filter with retained algae were placed on a microscope slide, embedded it with the anti-fading reagent Citifluor AF1 (Citifluor, London, UK), and covered with a cover slide for microscopic inspection (Olympus BX51FL, Olympus Corporation, Tokyo, Japan).

### 2.3. Sediment P Fractionation

The content of P fractionations in sediment were quantified through the EDTA sequential extraction procedure according to Golterman [34]. This method grouped sediment P into inorganic fractions including iron-bound P (Fe (OOH)~P), calcium bound P (CaCO3~P), and organic fractions including acid-soluble organic P (ASOP) and hot NaOH-extractable organic P ($P_{alk}$). P sequential fractionation extraction steps were as follows: First, extraction of P from Fe (OOH) was as follows: 1.5 g fresh sediment was extracted twice with 30 mL of 0.05 M Ca-EDTA for 2 h. Extraction of P from CaCO3 was carried out with sufficient repeated extractions using 30 mL of 0.1 M Na$_2$-EDTA for 17 h. Third,

for extraction from ASOP, an extraction with 30 mL of 0.5 M $H_2SO_4$ for 30 min was carried out. Fourth, extraction from NaOH-$P_{alk}$ consisted of 30 mL of 2 M NaOH at 90 °C for 30 min. The extracted concentrations of phosphate were measured by the blue molybdate method [28]. The dry weight of sediment was used to determine the content of different P fractions in sediment.

The significance of differences among variables before and after water diversion was determined via independent-sample t test [35,36]. Differences were considered significant at $P < 0.05$.

## 3. Results

### 3.1. Concentrations of Different Forms of P and N in Lake Yuehu and the Han River

Before water diversion, the average DIN concentration in the Han River was 2.38 mg·$L^{-1}$, which was four times higher than Lake Yuehu (Figure 2a). The DIN concentrations ranged from 0.329–1.595 mg·$L^{-1}$ in Lake Yuehu. $NO_3$-N was the main component of the DIN in all sampling sites, ranging from 0.134 to 0.757 mg·$L^{-1}$. It was notable that after water diversion, the N concentrations, especially in form of $NO_3$-N in Lake Yuehu, increased and were almost similar with that of the Han River on 28 July, with DIN values at 1.27 mg·$L^{-1}$ and 1.33 mg·$L^{-1}$, respectively (Figure 2b; Figure 3a). The DIN concentration sharply decreased on 7 August, but then returned to a higher level (Figure 3a). Additionally, there was no significant difference in N concentrations among the three sampling sites (Figure 3a).

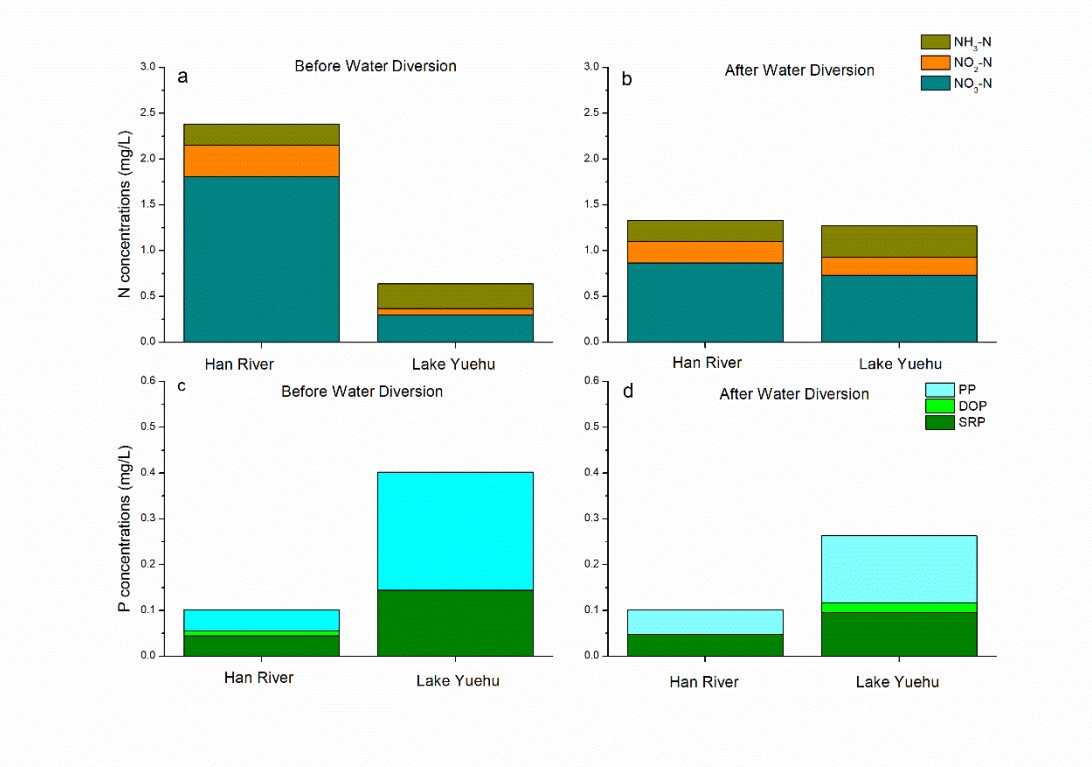

**Figure 2.** Concentrations of different forms of N and P of before and after water diversion in the Han River and Lake Yuehu. (**a**) Different forms of N before water diversion; (**b**) different forms of N after water diversion; (**c**) different forms of P before water diversion; (**d**) different forms of P after water diversion.

As for the P concentration, before water diversion, the concentrations of P—including TP, SRP and PP—in the Han River were obviously lower than in Lake Yuehu; the TP concentration was four times higher in Lake Yuehu than Han River (Figure 2c). In the water body of Lake Yuehu, the SRP concentrations ranged from 0.046–0.214 mg·$L^{-1}$. After water diversion, the concentration of TP decreased in Lake Yuehu, with its value significantly lower than before water diversion ($P < 0.05$), but little higher than the Han River (Figure 2d). The TP and SRP concentrations reached their lowest

levels on 28 July and 7 August, respectively. After that, the TP increased from 7 August and the SRP concentration also increased and got the peak to 0.2 mg·L$^{-1}$ on 14 August in Lake Yuehu (Figure 3b). In addition, no significant difference in P concentrations among the three sampling sites was observed during the study period (Figure 3b).

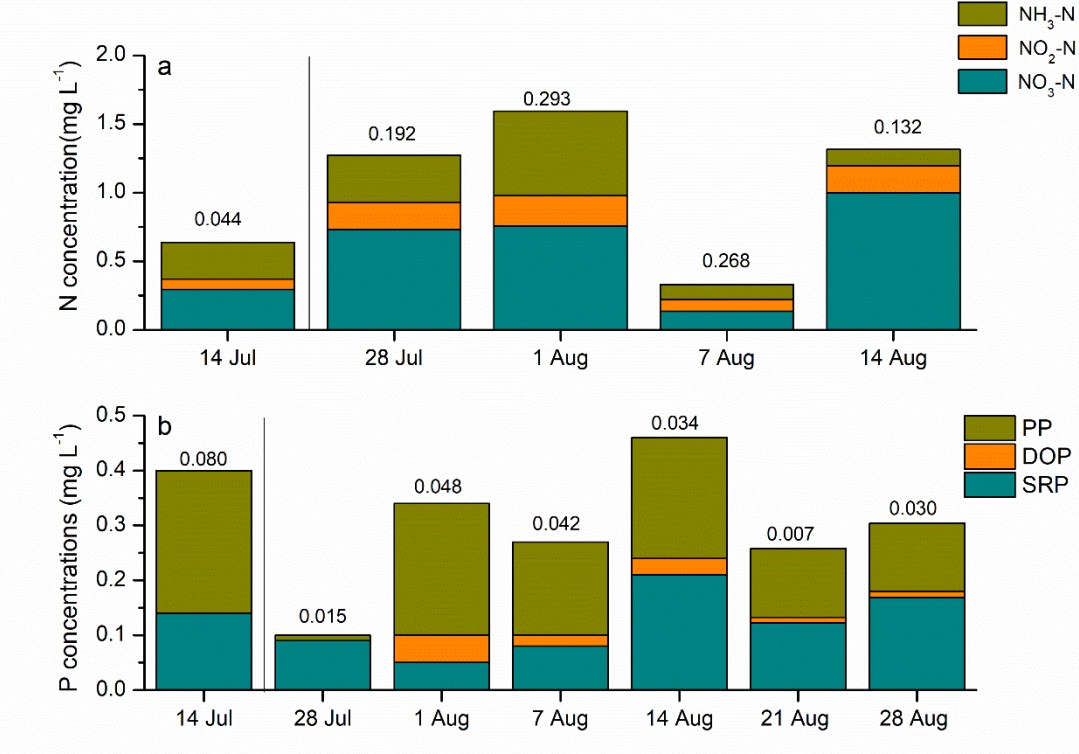

**Figure 3.** Concentrations of different forms of N (**a**) and P (**b**) in Lake Yuehu. The numbers above dates are the standard deviation of dissolved inorganic of N (DIN) and total P (TP) of the three sites. The left side of the line indicates the period before water diversion; the right side indicates the period after water diversion.

## 3.2. APA and Sediment P Fractionations before and after Water Diversion in Lake Yuehu

The average levels of APA in the water column were 1297.0 and 1109.1 µg·g$^{-1}$·h$^{-1}$ before and after water diversion, respectively. There were no significant differences among data from the three sampling sites in the lake, nor between the data from before and after the process of water diversion at the same site (Figure 4). As for the parameters of kinetics of alkaline phosphatase in sediment including $V_{max}$ and $K_m$, no significant differences were observed before or after the process of water diversion at the same site (Table 1).

**Table 1.** Kinetic parameters of alkaline phosphatase before and after water diversion.

| Water Diversion | $V_{max}$ (µmol·L$^{-1}$·h$^{-1}$) | $K_m$ (µmol·L$^{-1}$) |
|---|---|---|
| Before | 1299.752 | 0.133 |
| After | 1841.685 | 0.155 |

Before water diversion, the contents of Fe (OOH)~P), CaCO3~P, ASOP and P$_{alk}$ were 538.297, 390.840, 113.1767 and 28.920 mg·kg$^{-1}$, respectively, with Fe (OOH)~P) content the largest proportion. After the procedure, their values were 81.103, 103.263, 51.037 and 12.887 mg·kg$^{-1}$, respectively, which were significantly lower than those quantified before the process ($P < 0.05$). The average values of Fe (OOH)~P), CaCO3~P, ASOP and P$_{alk}$ were 6.64, 3.78, 2.22 and 2.24 times higher than after water diversion (Figure 5), among which Fe (OOH)~P) exhibited the highest value. In terms of

extracellular alkaline phosphatase determination at the single-cell level, ELF signals in Chlorophyta such as *Scenedesmus* could be observed (Figure 6a), while no ELF-labelling cell was observed in *Microcystis* during our study (Figure 6b).

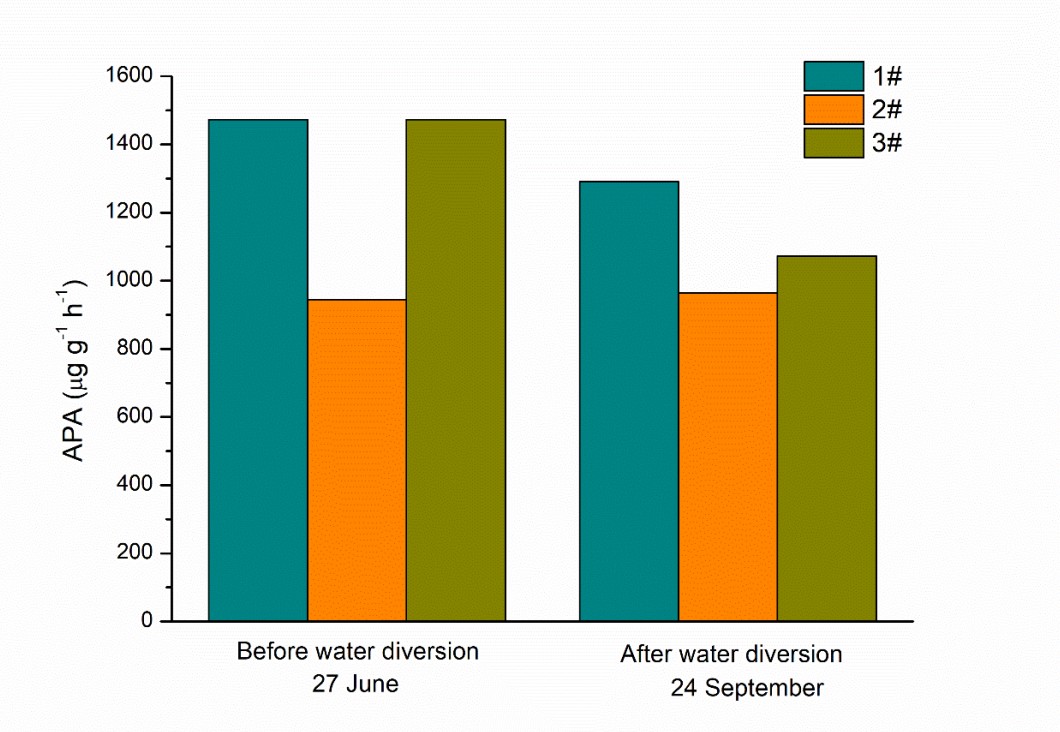

**Figure 4.** Alkaline phosphatase activity in the water column before and after water diversion in Lake Yuehu.

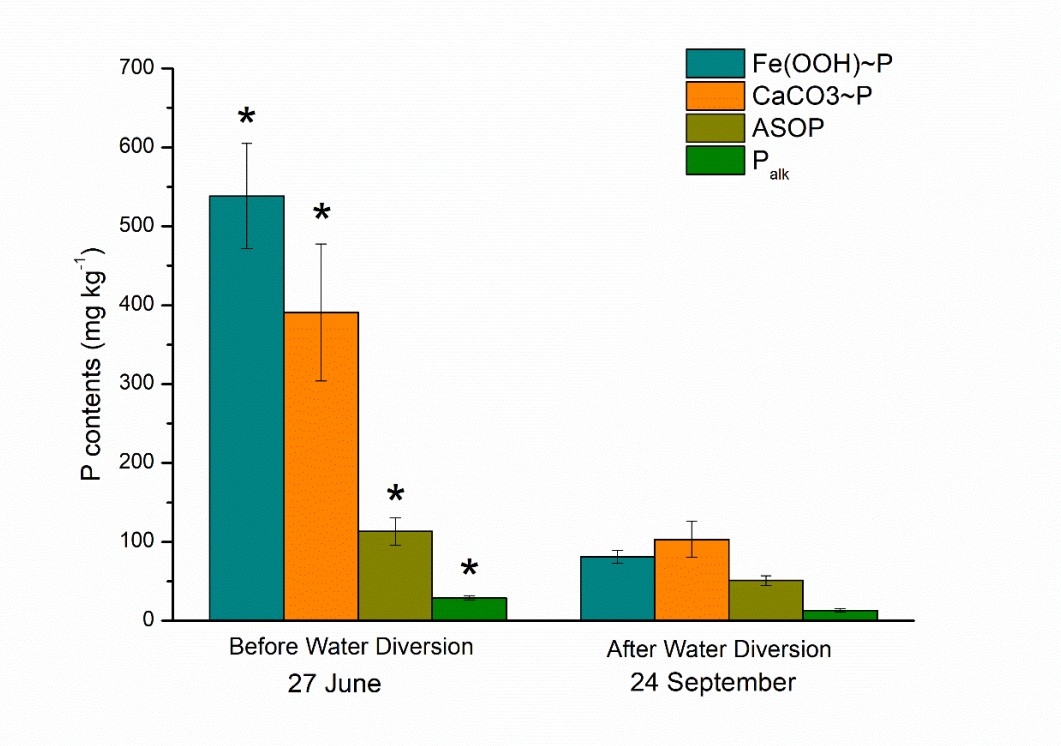

**Figure 5.** Sediment P fractionations before and after water diversion in Lake Yuehu. Asterisks (*) indicate significant difference ($P < 0.05$).

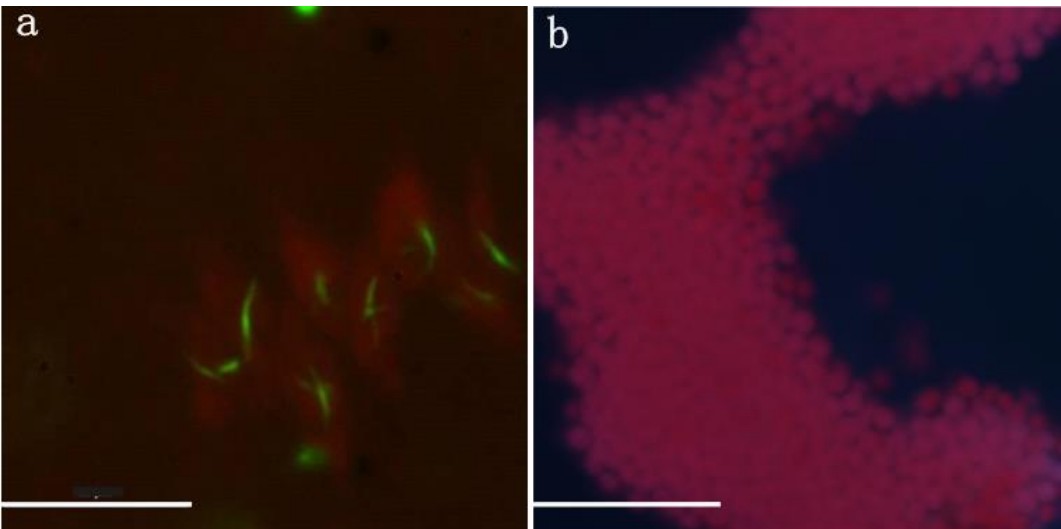

**Figure 6.** Enzyme-labeled fluorescence (ELF) detection in fluorescence microscope in field sample. (**a**) *Scenedesmus* (Chlorophyta); (**b**) *Microcystis* (Cyanophyta). Red color indicates chlorophyll fluorescence and green-yellow fluorescence marks sites where extracellular alkaline phosphatase functioned. Scale bar represents 20 μm.

### 3.3. The Composition of Algal Density in Lake Yuehu

Before water diversion, algal taxa were diverse in Lake Yuehu, containing Cyanophyta, Bacillariophyta, Cryptomonas, Dinoflagellate, Euglenophyta and Chlorophyta. The cell density of total phytoplankton, Cyanophyta and *Microcystis* were $2.42 \times 10^7$, $9.43 \times 10^5$, and $3.63 \times 10^5$ cells $L^{-1}$, respectively on 14 July. The *Microcystis* accounted for 38.46% of Cyanophyta and only 1.50% of total phytoplankton. After water diversion, the phytoplankton was mainly composed of *Microcystis* and its cell density increased steeply to $6.03 \times 10^7$ cells $L^{-1}$ on 28 July, which accounted for 59.14% of total cell density. The percentages of the cell density of *Microcystis* to total phytoplankton increased sharply to 60–90% at different sampling sites after water diversion (Figure 7).

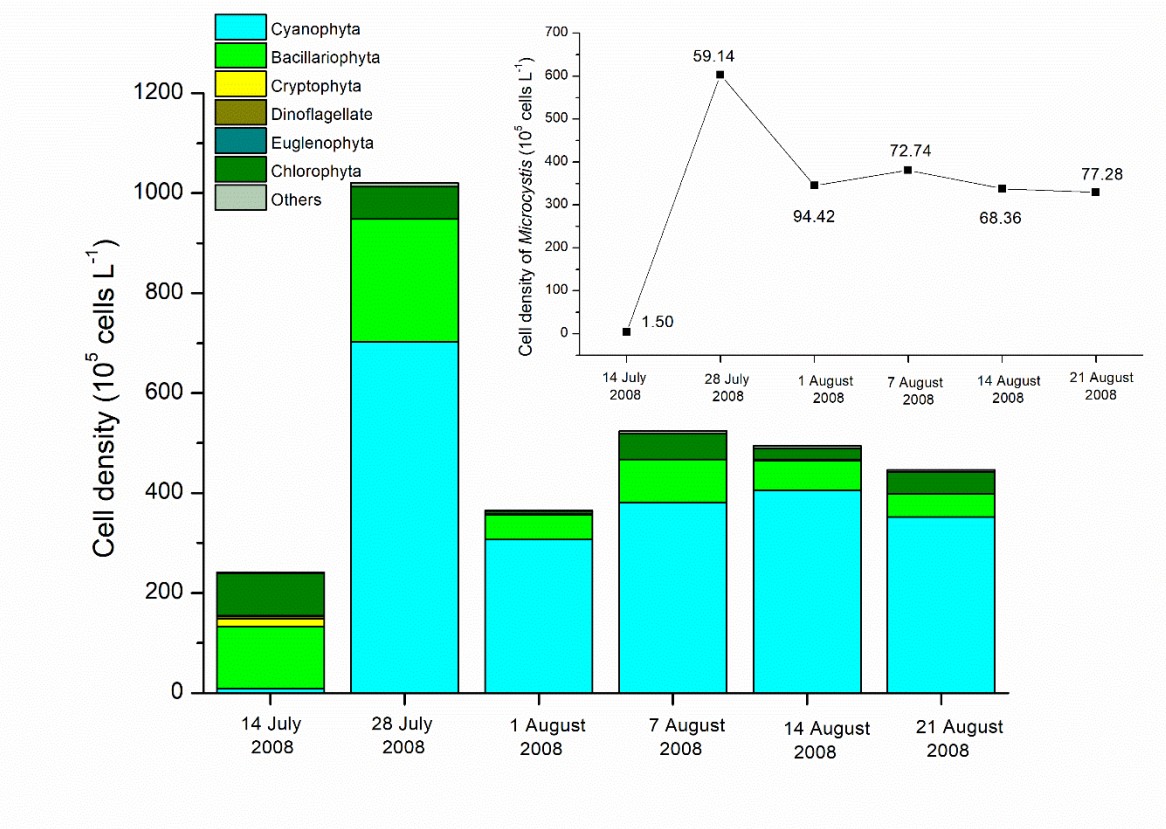

**Figure 7.** Phytoplankton composition and cell densities in Lake Yuehu during the study period. Numbers besides the dots in the figure indicate the percentages of the ycell density of *Microcystis* to total phytoplankton (%).

## 4. Discussion

Water diversion is proposed as an effective measure for lake restoration, since its active effects have been shown to improve water quality and alleviate algal blooms all over the world [37–40]. However, negative conflicting impacts have been presented in some cases [17,19,22]. In the present study, a *Microcystis* bloom occurred just after water diversion in the studied lake. How water diversion contributed to the cyanobacterial bloom remained unclear. A track investigation on nutrient fluctuations and the dominant phytoplankton response to ambient P variation during the diversion in the field was conducted to illustrate the mechanism of *Microcystis* bloom in Lake Yuehu after water diversion from the Han River. The *Microcystis* bloom was due to the coupling of the nutrient supply and the nutrient uptake characteristics of *Microcystis*. A pulse supply of N carried from the source and the large amount of P released from the sediment may satisfy the need of *Microcystis* and triggered its bloom. Our results indicate that the short-term water diversion from the Han River to the urban Lake Yuehu had a negative effect in alleviating the cyanobacterial bloom.

The concentrations of different forms of N and P varied greatly in Lake Yuehu after water transfer. Spatially, there was no significant difference among the sampling sites in terms of N and P concentrations before and after water diversion. The increases of bioavailable $NO_3$-N and $NH_4$-N, might be due to the water from the Han River with a high N loading (Figure 2a,b). On the contrary, the P content had been diluted by the water from the river with lower P concentrations (Figure 2c,d). Thus, water diversion could positively affect lakes by diluting the nutrients concentration [11–13,41], but also could negatively affect them if water transfer directly brings large amount of nutrients to the lake. For example, water transfers from the Yangtze River to Lake Taihu in 2007 added more nutrients of N and P to the lake in the spring and summer [17]. Water introduction might change the nutrient regime by changing water–sediment exchange. In Lake Yuehu, the TP concentration increased after

water transfer, while the SRP concentration still decreased and then increased its peak on 14 August (Figure 3b). The internal P release might have contributed to this increase. The different contents of inorganic P forms in sediment were significantly decreased after the process of water diversion, with the highest difference in Fe (OOH)~P content (Figure 5). Phosphate was preferentially bound on the Fe (OOH) particles [42] and apt to be released upon dissolution of ferric oxides under strong reduction conditions [43]. The water introduction to Lake Yuehu created anaerobic conditions at the water–sediment interface, which can trigger inorganic P releasing into the water column from the sediment [44,45]. Opposite to the inorganic P fractions in sediment, the organic ones including ASOP and $P_{alk}$ showed no significant difference (Figure 5), nor were the kinetic parameters of alkaline phosphatase ($V_{max}$, $K_m$) hydrolyzing organic P in sediment (Table 1), indicating that the elevation of TP concentration—especially the SRP in the water column after water diversion—was primarily attributable to the release of inorganic P fractions rather than the biological process mediated by alkaline phosphatase in the sediment. This process could also be excluded in the water column of Lake Yuehu, although one of the internal sources of bio-available P in the water column is phosphomonoesters hydrolysis by alkaline phosphatase to phosphate [46]. No significant difference of APA in the water column before and after water diversion was observed (Figure 4), which was consistent with previous studies [24,25]. Briefly, water transfer might dilute P content but enhance N concentration by bringing a high N source or triggering internal P release in the received lakes. The internal P release always took place due to the concentration gradient caused by water with low levels of P [47].

The algal taxa were diverse in Lake Yuehu before water diversion but after water diversion *Microcystis* dominated and formed a surface bloom (Figure 7). This unexpected cyanobacterial bloom might be a consequence of the response of *Microcystis* to N and P variation in terms of content and supply mode. Similarly, the transition of the phytoplankton community and alteration of the predominant species are induced by the influent water with pulsed nutrients, which could subsequently accelerate algal blooms [16,21,22]. N was always a key limiting factor for the growth of the non-$N_2$ fixing cyanobacteria in freshwater [48–50]. *Microcystis* blooms in Western Lake Erie responded quickly and strongly to N inputs [51,52]. Before the procedure of water diversion, N concentration was clearly lower in Lake Yuehu. The increase of bioavailable N transferring from the Han River might have relieved the N limitation suffered by *Microcystis* and drove its growth and reproduction significantly in Lake Yuehu. Meanwhile, the P-using strategy of *Microcystis* might play an important role in its dominance. After the water transfer, the SRP concentration was decreased in Lake Yuehu on 28 July, when the *Microcystis* bloom occurred (Figure 3b; Figure 7). *Microcystis* might not suffer from a P limitation at this time since it did not produce extracellular phosphatase, as evidenced by in situ ELF labeling, which was an indicator of P deficiency (Figure 6). Our previous study revealed that *Microcystis* spp. displayed a competitive advantage at low P concentrations, with different strategies to use P such as rapidly uptaking and storing inorganic P, while also elevating the P deprivation of the coexisting phytoplankton species [53]. In addition, with the bio-available P pulse released from the sediment, the SRP concentration in the water column also increased, which probably supported the maintenance of the *Microcystis* bloom. Our findings are consistent with another study where, although water quality was remarkably improved due to dilution by low-nutrient Columbia River Water in Lake Moses, more TP and cyanobacteria were attributed to greater internal TP loading [41]. Internal P sources contribute a lot to the triggering of algal blooms in surface water systems [54,55]. We proposed that the transformation of nutrients in Lake Yuehu due to water diversion have might triggered the *Microcystis* bloom, since no annual *Microcystis* bloom was observed before the water diversion at the same season with similar meteorological conditions.

## 5. Conclusions

The most notable observation in the current study was the unexpected cyanobacterial bloom occurring promptly after introducing the water from a river to alleviate eutrophication in Lake Yuehu. Therefore, we hypothesized that the nutrient variation in the lake met the growth requirement of

*Microcystis* and exploited its advantages in scavenging P, which was indicated by the field tracing investigation. Although a large amount of bioavailable N originating from source water in the Han River might alleviate the N limitation of the non-N fixing cyanobacteria *Microcystis*, the high adaptation to low P concentration may play an important role in its dominance. Furthermore, the P concentration increased, which increased due to the anaerobic-pulsed release from the sediment, probably maintaining the growth and proliferation of *Microcystis*. Thus, water transfer would increase the risk in terms of elevated N and P concentrations and cyanobacterial bloom. We must therefore exercise caution when conducting water transfer projects.

**Author Contributions:** X.C. (Xiuyun Cao), Y.Z. and C.S., conceived and initiated the project. X.C. (Xiuyun Cao) and C.S. took field samples and chemical analysis. X.C. (Xiaoyan Chen), D.B., Y.Z., X.C. (Xiuyun Cao), and C.S. analyzed the data and composed the manuscript. All authors analyzed the results and edited the manuscript. All authors have read and agreed to the published version of the manuscript.

**Funding:** This work was supported by grants from National Key Research & Development Program of China (2016YFE0202100), the National Natural Science Foundation of China (41877381, 41273089), the National Major Science and Technology Program for Water Pollution Control and Treatment (2017ZX07603) and the State Key Laboratory of Freshwater Ecology and Biotechnology (2019FBZ01).

**Conflicts of Interest:** The authors declare that they have no conflicts of interests.

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
