# Peer review of "Microcystis Bloom in an Urban Lake after River Water Diversion—A Case Study"

_water, doi:10.3390/w12061811_

Round 1

Reviewer 1 Report

This study with title of ‘Microcystis bloom in an urban lake after river water diversion: a case study’ is generally not well-written. In my opinion the paper has no shortcomings regarding some data analyses and text. Below I have provided some remarks on the text as it is often vague and long-winded. Given these shortcomings the manuscript requires major revisions.

Major comments

  • Section of abstract should be reconstructed, and authors should mention a brief of methodology, results and conclusion in the abstract.
  • Please provide some basic information about geometry and hydro climate of lake and river
  • Keywords should be revised fundamentally, please note that the words mentioned in the title, aren’t allowed in the keywords.
  • The novelty of the study it is not clear, and it should be clarify at the end of “introduction” by authors.
  • The manuscript mainly has been developed based on water diversion from Han River to Lake Yuehu, please provide an explanation regarding to this diversion, how and how much water transfer to lake by which scheduling.
  • In the section of the introduction, the authors need to go much further in talking about the importance study of the microcystis over the world and in the study area.
  • It would be better to restructure the introduction section in a better way. Three paragraphs are enough: i) First one for the importance of the issue + challenges, ii) Second one for literature review, and iii) Third one for objectives and novelty of the study (it is more important).
  • The writing of the manuscript it’s not academic, it should be revised.
  • Some sentences in the manuscript is too long and they should be rewrite and summarize.
  • There are some grammatical problems and the writing of the manuscript, it should be revised by the native speaker.
  • In the discussion section, some repetitive sentences from the method and results (Lines 231-233) are presented. Therefore, the authors should revise the discussion section fundamentally.
  • The conclusion section should be improved by the authors.
  • In the Figure 1(Location of Han River and Lake Yuehu.), authors should add better scale bar and north arrow. Provide also legend regarding black triangles, and location in China
  • In the section of the “2.4. Statistical analysis” it is more important to referee some past references which applied the Statistical analysis such as independent-sample t test.
  1. Haghighi, A. T., Darabi, H., Shahedi, K., Solaimani, K., & Kløve, B. (2020). A scenario-based approach for assessing the hydrological impacts of land use and climate change in the Marboreh Watershed, Iran. Environmental Modeling & Assessment, 25(1), 41-57.
  2. Pirnia, A., Darabi, H., Choubin, B., Omidvar, E., Onyutha, C., & Haghighi, A. T. (2019). Contribution of climatic variability and human activities to stream flow changes in the Haraz River basin, northern Iran. Journal of Hydro-environment Research, 25, 12-24.
  • Please correct the name of lake somewhere in the manuscript it is called Lake Yue (e.g. Line 78)
  • Please provide a discussion on uncertainty of data, How 3 sample points is enough for this study.

Reviewer 2 Report

Main comments:

  • More details on chemical analysis should be given. Methods of analysis should be provided.
  • Explain in how many repetitions chemical analyses of water and alkaline phosphatase activities and the kinetics were performed. The same should be done for sediment P fractionation analyses.
  • Please provide details of sampling dates for chemical analyses for the period before and after water diversion.
  • The Statistical analysis section should not be distinguished.
  • It should be clarified whether water samples have also been collected from the Han River. Complete the description and indicate in Figure 1.
  • It should be clarified whether "all the sampling times" is the same as before water diversion (24th June to 14th July, 2008). How many samples were collected during the period from 24th June to 14th July, 2008.
  • Explain how the N and P concentrations at sampling points 1, 2 and 3 differed during the period before and after water diversion.
  • In Figure 3 the dates before and after water diversion should be marked.
  • Explain why N-concentration were not carried out on 21th and 28th August
  • Were other measurements relevant for Microcystis bloom analysis carried out, such as: water temperature, dissolved oxygen content, pH, etc.?
  • What meteorological conditions (air temperature, wind speed, sunlight, etc.) occurred during the research period?
  • The results presentation should be extended.

Specific comments:

  • Line 21 Explain the meaning of the abbreviation ELF
  • Line 78. Change Yue to Yuehu
  • Line 90. Explain the meaning of the abbreviation APA
  • Line 193 – The authors indicate that “Before water diversion, the algal taxa were diverse in the Lake Yuehu, containing Cyanophyta, Bacillariophyta, Euglenophyta, Pyrrophyta, Cryptomonas, Xanthophyta and Chlorophyta in Figure 6 is also Dinoflagellate? In addition, a change in the type of graph should be considered (Figure 7). It should be clarified whether 14th July Cyanophyta were in water or not?

Round 2

Reviewer 1 Report

No Further comments, I recommend to be accepted,

Reviewer 2 Report

The authors responded to all comments and suggestions contained in the review.